# The Missing “lnc” between Genetics and Cardiac Disease

**DOI:** 10.3390/ncrna6010003

**Published:** 2020-01-14

**Authors:** Maral Azodi, Rick Kamps, Stephane Heymans, Emma Louise Robinson

**Affiliations:** 1INRS Centre Armand-Frappier Santé Biotechnologie, Laval, QC H7V 1B7, Canada; azodimaral@gmail.com; 2Department of Genetics & Cell Biology, School for Mental Health and Neuroscience (MHeNS), Maastricht University, P.O. Box 616, 6200 MD Maastricht, The Netherlands; rick.kamps@maastrichtuniversity.nl; 3Department of Cardiology, Cardiovascular Research Institute Maastricht (CARIM), Maastricht University, P.O. Box 616, 6200 MD Maastricht, The Netherlands; s.heymans@maastrichtuniversity.nl; 4Centre for Molecular and Vascular Biology (CMVB), Department of Cardiovascular Sciences, KU Leuven, B3000 Leuven, Belgium

**Keywords:** cardiovascular disease (CVD), long non-coding RNA (lncRNA), whole exome sequencing (WES), whole genome sequencing (WGS), pathogenic gene variants

## Abstract

Cardiovascular disease (CVD) is one of the biggest threats to public health worldwide. Identifying key genetic contributors to CVD enables clinicians to assess the most effective treatment course and prognosis, as well as potentially inform family members. This often involves either whole exome sequencing (WES) or targeted panel analysis of known pathogenic genes. In the future, tailored or personalized therapeutic strategies may be implemented, such as gene therapy. With the recent revolution in deep sequencing technologies, we know that up to 90% of the human genome is transcribed, despite only 2% of the 6 billion DNA bases coding for proteins. The long non-coding RNA (lncRNA) “genes” make up an important and significant fraction of this “dark matter” of the genome. We highlight how, despite lncRNA genes exceeding that of classical protein-coding genes by number, the “non-coding” human genome is neglected when looking for genetic components of disease. WES platforms and pathogenic gene panels still do not cover even characterized lncRNA genes that are functionally involved in the pathophysiology of CVD. We suggest that the importance of lncRNAs in disease causation and progression be taken as seriously as that of pathogenic protein variants and mutations, and that this is maybe a new area of attention for clinical geneticists.

On completion of the human genome in 2001, it was an unprecedented shock to find that less than a quarter of the number of genes were present from the original prediction at 100,000 or so [1,2]. This finding left researchers with the conclusion that the remainder of the human genome is “junk DNA”, coined genomic “dark matter”. The revolution in deep sequencing RNA technologies of the last 15 years has revealed that, whilst only 2% of the human genome encodes classical protein-coding genes, nearly 90% of the entire human genome is transcribed into RNA, not constitutively but in a highly spatially (cell- and tissue-type specific) and temporally (stage in differentiation and maturity or response to different environmental cues) specific manner [3]. These regions of the genome have now been re-named as the “non-coding genome.” The majority of the non-coding genome is comprised of long non-coding RNAs (lncRNAs), defined as functional RNA molecules longer than 200 nucleotides that do not possess protein-coding capacity.

According to the latest GENCODE database, there are 19,965 protein-coding genes and 17,910 lncRNA genes [4]. Like protein-coding genes, lncRNAs have different isoforms and splice variants. It has been estimated that up to 270,044 different lncRNA transcripts can be expressed from the human genome [5].

Up to 80% of these remain uncharacterized. There are a number reasons holding back the full characterization of the entire non-coding genome, including low copy numbers (more sensitive detection methods needed), cell-type or sub-type specificity (maybe be undetectable in whole tissue samples), poor conservation and non-canonical gene structures and features. Importantly, nearly 100,000 of all lncRNA transcripts are disease-associated in their expression pattern [5].

The functions of lncRNAs is as varied as that of proteins and includes, but is not limited to, acting as guide molecules for epigenetic and chromatin remodelers in the nucleus, direct antisense inhibitor/disruption of transcription, co-factors and regulators of proteins inside and out the nucleus and in transport [6]. The original dogma of molecular biology describes that only proteins encoded by genes (exons/mRNA) exert biological functions but nowadays it is evident that also RNA transcripts, particularly lncRNAs, exert important regulatory roles in cell physiology and intercellular communication [7]. Thus, the classical central dogma in molecular biology has been revisited as of the last decade (Figure 1).

CVD is the leading cause of mortality and morbidity globally and is responsible for around 47% of deaths worldwide. Among the various types of CVD that can affect both blood vessels and the heart include cardiomyopathies, coronary artery disease, heart failure and arrhythmias. Up to 85% of CVD-related deaths are due to heart failure and an eventual heart attack (myocardial infarction). Heart failure is the stiffening or weakening of the heart muscle to the extent that cardiovascular demands of the head and body are no longer met. Heart failure currently affects 2% of the adult population, rising to 10% in those over 65 years of age [8]. With the percentage of the global population over 65 years of age predicted to reach 30% by 2030, the socioeconomic burden of CVD is set to escalate [9].

LncRNAs are emerging as key regulators of ageing and CVD pathology. Many have been identified to regulate gene expression through epigenetic mechanisms, splicing modulation, chromatin remodeling or microRNA interaction, cofactors to regulate key proteins in cells of the cardiovascular system as well as potential biomarkers in the circulation for CVD diagnosis and prognosis [10].

Myosin heavy-chain-associated RNA transcripts (*Myheart* or *MHRT*) is a lncRNA cluster, which is overlapping and antisense to the myosin heavy chain 7 (Myh7) gene locus. MHRT protects the heart from pathological hypertrophic remodeling by suppressing the activity of immediate early stress gene-associated chromatin remodeling factor BRG1 in cardiomyocytes [11]. *MHRT* itself is activated in response to stress by the Brg1–HDAC–PARP complex, signifying a self-regulatory and protective feedback loop. However, downregulation of MHRT has been observed in cardiac tissue in a number of cardiac pathologies [11]. Differential methylation and expression of *MHRT* has also been implicated in underlying sex differences in left ventricular cardiac remodeling, through methyl CpG binding protein 2 and pri-miR-208b [12]. 

Metastasis associated lung adenocarcinoma transcript-1 (*MALAT-1*) is a 8 kb well-defined, conserved and highly abundant lncRNA in many cell types in higher organisms and has been implicated in a number of disease entities [13]. Its expression in elevated in the heart and cardiovascular system in many different pathologies. *Malat-1* deficiency in immune cells promotes atherosclerosis in ApoE-/- mice and is thus protective against atherosclerosis [14]. In addition, *Malat-1* has been shown to positively regulate cardiac fibrosis through sponging and sequestering microRNA-145 in myocardial infarction, promoting fibroblast proliferation, collagen production and α-SMA expression in cardiac fibroblasts [15].

Cardiac-hypertrophy-associated epigenetic regulator (*Chaer*) is a novel immediate early gene elevated in response to neurohumoral hypertrophic gene stimulation (endothelin-1/angiotensin II) [16]. It acts as an epigenetic modulator, by binding and suppressing the Ezh2 catalytic unit of the polycomb repressor complex 2, preventing H3K27 methylation at key hypertrophy-associated genes, including *NPPA*.

LncRNAs can also play a role in cell–cell communication. Whilst primarily expressed in cardiofibroblasts, myocardial infarction-associated transcript1 (*Mirt1*) plays a role in the cardiomyocyte survival and the acute inflammatory regulation after myocardial infarction [17,18]. Knockdown of *Mirt1* in murine cardiomyocytes and fibroblasts attenuated nuclear transport of NF-κB and expression of the pro-inflammatory cytokines IL-6, IL-1β and TNF-α, as well as cardiomyocyte apoptosis in acute myocardial injury. These are just some examples of the vital regulatory roles of lncRNAs in CVD mechanisms [19].

Both genetics—that is the DNA sequence that an individual inherits—as well as environmental factors play a role in CVD risk. These factors interact and intersect in a complex manner.

It is thought that genetic background contributes to about half of the heart disease risk (i.e., vascular, cardiomyopathies, electrophysiological properties of cardiomyocytes, ion transportation and congenital heart disease) [20]. To reduce the risk of CVD up to 50%, we can fight against traditional lifestyle-related risk factors (e.g., smoking, obesity, hypertension, high cholesterol and diabetes). However, these associated causes contribute to a fraction of CVD causation and progression, which differs between the particular form of CVD and individual genetic background.

Hypertrophic cardiomyopathy (HCM, 1:500), dilated cardiomyopathy (DCM, 1:2500), arrhythmogenic cardiomyopathy (ACM, 1:5000) and restrictive and non-compaction cardiomyopathy are the most common type of genetic cardiomyopathies [21].

Genetic mutations in more than 30 genes have been found in familial DCM. The majority of protein-coding gene variants and mutations associated with DCM encode key components of the sarcomere or cytoskeleton of cardiomyocytes [22].

For example, approximately 20% of cases of familial DCM happens in mutations in one gene—*TTN*. The *TTN* gene provides instructions for making the protein titin, which provides structure, flexibility and stability to sarcomeres. The *TTN* gene also plays a role in chemical signaling and in assembling new sarcomeres [23].

Coronary artery disease (CAD) can be a heritable disorder for which there are more than 60 genetic loci associated; however, they account for only 10% of disease heritability and only 33% of these loci were associated with traditional CAD risk factors [24]. Whole-exome sequencing may also discover rare genetic variants that actually protect against coronary artery disease. Rare variant association studies indicated that there are inactivating mutations in at least nine genes with risk of CAD [24].

To tailor the medical treatment course to the individual characteristics and etiology of disease of each patient, we will rely on our understanding of how a person’s unique molecular and genetic profile makes them susceptible to certain diseases [25]. Tailored medical treatment opens new horizons in modern molecular medicine. A patient’s genetic (and epigenetic) profile increases our ability to predict the most beneficial medical treatment by eliminating ineffective treatments. As CVDs represent a major economic burden on health care systems, set to increase with the ageing global population, searching for a disease management strategy such as tailored medical treatment will be necessary in this area [26].

In the last decade, next generation deep sequencing (NGS) technology has started the paradigm shift in the search for underlying disease-causing variant reliability and classification in routine clinical cardiovascular practice. A number of clinical NGS applications are utilized, including variant detection in autosomal dominant cardiogenic disease based on DNA-sequencing, acquired or somatic variant analysis caused by environmental factors, identifying risk modifiers as other genetic factors, detection of spliceogenic variants based on RNA-sequencing (RNA-seq) and as a biomarker application for pharmacogenetics or drastic lifestyle changes.

Traditional Sanger sequencing is still performed to analyse specific DNA regions for heterozygous disease-causing variants of known cardiogenic genes or as a second independent technology confirming potential candidate NGS variants. Currently, the use of NGS almost replaced conventional Sanger sequencing or targeted sequencing and is a very common versatile approach for several clinical and non-clinical applications [27].

Different DNA approaches can be used according to the needs and the questions being addressed. DNA material can be enriched to analyze a limited number of target genetic regions, such as autosomal dominant cardio disease gene-panels or whole exome sequencing (WES) of all coding exons of the human genome. Alternatively, the complete genomic DNA can also be sequenced (whole genome sequencing, WGS). However, the use of WGS routinely in clinical genetics is limited by its financial cost and complexity of data interpretation, especially in expanding our understanding of the function of the non-coding part of the genome.

Previously, genome-wide association studies (GWAS) were used to determine whether any specific areas of the genome are associated with heart disease in large cohorts. These studies have rarely conclusively identified genes that underlie differences in heart disease due to genetic heterogeneity and the complexity of multi-gene–environment interactions [28].

Importantly, whole genome approaches such as WGS does not exclude non-coding regions, characterized or not. With lncRNAs emerging as having similarly important and diverse functions as proteins, variants and mutations in ncRNAs could reveal the genetic component of CVD that we currently do not understand or describe as idiopathic.

Nowadays, the identification of disease-causing variants using RNA-seq or transcriptomics is emerging as a clinical transcriptome profiling system. Splicing, the process of removal of the introns in the pre-mRNA molecule, is highly regulated by the RNA-splicing machinery and depends on specific genetic sequences to mark intron/exon junctions. Additionally, epigenetic factors (chromatin conformation and histone modifications) have also been implicated in the regulation of splicing [29]. Incorrect splicing, which occurs due to the presence of genetic variants, mostly in intronic regions, has been implicated as a cause of genetic disease. Targeted-sequencing or WES analysis is mainly limited to the protein-coding region (exons) of the genome and to a few intronic nucleotides, which are the most conserved splice sites adjacent to the exon and do not cover the remaining of the introns, promoter regions and non-coding RNAs, which are important for gene-expression regulation. In addition, computer-based tools and pipeline analyses are less accurate in predicting the effect of DNA-variants on gene-expression or RNA splicing. Therefore, mutations with a potentially pathogenic effect on RNA expression or processing are either not included in the regions of interest during WES sequencing or are missed by bioinformatics analysis [30]. Sequencing the transcriptome by RNA-seq is therefore a valuable approach to detect variants affecting RNA amounts, including mutations in transcription binding sites or promotors, or splicing in tissues where the gene is expressed. RNA-seq allows the analysis of the transcriptome at an unprecedented depth [31].

Recently an interesting study on lncRNA and coding RNA profiling using strand-specific RNA-Seq in 28 human HCM patients revealed interesting data in processes between dysregulating coding genes and lncRNA genes in HCM versus healthy control conditions [32]. Although, combining RNA-Seq with WGS protocols and integrated analyses will also aid identification of RNA-editing events and will increase the chance of finding the causative DNA variant. RNA-Seq also has some limitations. RNA is less stable than DNA and requires higher care in handling and storage of the samples. In addition, RNA expression (and processing) is tissue-dependent and the tissue available for analysis may not express the gene with the defect. One of the first studies has shown the benefit of RNA-Seq in genetic diagnosis, yielding a 35% overall diagnosis rate for rare muscular disorders in screening splice variants in whole transcriptomic data [33].

A perfect example of how a whole genome approach reveals the importance of non-coding genetic components in CVD is that of the discovery of *MIAT* (myocardial infarction-associated transcript [34]). In 2006, a large-scale single nucleotide polymorphism (SNP) association study reported that in the Japanese population, there are six SNPs in the MIAT locus, a previously unidentified long intergenic non-coding RNA.

Yet, whilst some basic mechanisms of *MIAT* have been investigated mostly in in vitro systems, the pathogenicity or pathobiology of *MIAT* variants has not been investigated further in patients or larger and broader population studies, nor is it included in the targeted panel of cardio pathogenic genes in the clinic [35,36].

## Future Perspectives

An array of different methods and technologies are now available to enable us to identify the genetic basis of disease, with NGS becoming the prime diagnostic test. NGS has brought unprecedented advances in understanding the biology of diseases, with important clinical implications. The costs of short-read sequencing have become extremely low. Nowadays, WES/WGS and RNA-seq are becoming routine in clinical practice for genetic disorders, but the eventual step in diagnostics in the years to come will be WGS/RNA-seq by short-read sequencing and later by hybrid sequencing, involving long-read sequencing for structural variants and repeat sequences.

Technical breakthroughs and increased bioinformatics power and sophistication make NGS technology increasingly more powerful. It is crucial that these advances are accompanied by increasing awareness of its strong potential by physicians and patients. It is also of fundamental importance that the progress is paralleled by strict monitoring of the use of these technologies in relation to ethical issues, especially as NGS will in future not only be performed to identify the cause of disease, but also in the form of a personal genome to guide the life of a healthy person.

As the “non-coding” genome emerges as a complex network of molecular mediators in disease, we see lncRNA genes being included in classical pathogenic gene panels and whole “exome” sequencing to include mature lncRNA sequence. With the advent of faster and cheaper next generation sequencing, and the $1000 human genome well superseded, whole genome approaches should be used where possible in clinical genetics to avoid omitting genomic “dark” matter, which may contain currently unknown regulatory regions [37].

The chronological lagging behind in revealing the existence of lncRNAs as important biological mediators, their functional characterization and lack of widespread education and knowledge transfer of their significance to the current medical community all contribute to the slow progress in incorporating lncRNAs in mainstream clinical genetic analyses.

With the molecular and chemical nature of non-coding RNAs—unique sequences of RNA nucleotides—lending themselves to specific and sensitive pharmacological inhibition using antisense technology, as well as the first RNA therapy now U.S. FDA approved (Onpattro, patisiran), we look forward to seeing the future of personalized RNA therapy in the treatment of CVD [38,39].

## Figures and Tables

**Figure 1 ncrna-06-00003-f001:**
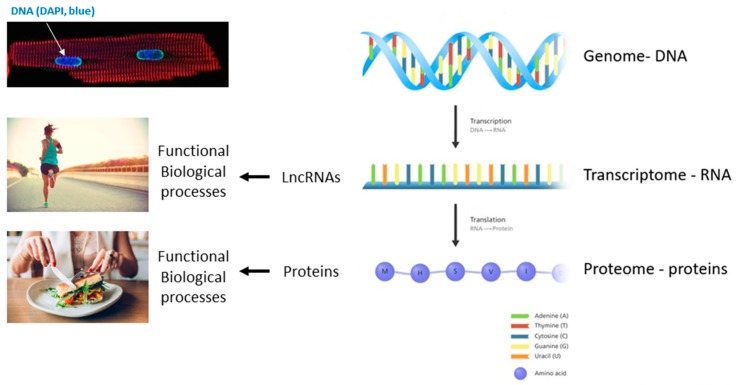
The revised central dogma in molecular biology. Image credit: this figure is adapted from Genome Research Limited.

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
