# Peer review of "The Missing “lnc” between Genetics and Cardiac Disease"

_ncrna, 2020, doi:10.3390/ncrna6010003_

Round 1

Reviewer 1 Report

The manuscript: „ The missing ‘lnc’ between genetics and cardiac disease” by Azodi et al. is a well written commentary on the hot topic of current cardiovascular research pointing to emerging need to translate knowledge on the long non-coding RNAs associated with cardiovascular disease to clinical praxis for individual patient´s benefit. Paper brings important and original view on the given topic and could be published in NcRNA journal after minor revisions.

Comments:

There is quite a high number of typing errors in the manuscript, thus paper should be carefully checked for them, e.g.: Line 4, author´s name “Azod i” should be corrected to “Azodi” Lines 22-28, bigger font size used as compared to the rest of manuscript Line 62, thete is double “Up to up to” Line 75 “remodelling” replace by “remodeling” as US English is used in whole manuscript (including word “remodeling” in other parts of paper) Line 95 “LnRNAs” replace by “LncRNAs” Line 99 “cardomyocyte” replace by “cardiomyocyte” Line 106 “Vascular” should not start with capital letter “V” Line 106 “electrophysiologial” replace by “electrophysiological” Lines 107, 117, please unify “%” or “percent” Line 118 insert space between “titin” and “which” Line 136 “utilised” replace by “utilized” (US English) Line 148 “analyse” replace by “analyze” (US English) Line 186 insert space between “defect.” and “One” Abbreviations: please, re-check if all abbreviations are properly explained when first time occur in the text (e.g. BRG1, MeCP2, HDAC, PARP, CAD, SNP…etc. are not explained). Alternativelly you may create the List of abbreviations and insert into the paper. Finally, please always use abbreviations when once were intorduced, e.g. use “CVD” instead of “cardiovascular disease” in lines 68, 221. I suggest to add explaining text to the Figure 1, such as “Original dogma of MB expected that only proteins encoded by genes (exons) exert biological functions, but nowadays it is evidenced that also RNA transcripts/molecules encoded by introns, particularly miRNAs and lncRNAs exert important biological functions, especially regulatory roles in cell physiology” or something similar to make the Figure self-explaining without reading whole paper. Check the text for fluent English and English grammar, especially in Future directions some sentences are not clear in meaning, e.g.: Line 198, first sentence, subject is missing: WHO?? is currently available to identify…? Maybe WE??. Alternatively, exclude words “With the” and start sentence with “Different methods…” Line 205, the same as previous – exclude “With the.” or add the subject to the sentence

Author Response

*our responses are marked with an asterisk.

The manuscript: „ The missing ‘lnc’ between genetics and cardiac disease” by Azodi et al. is a well written commentary on the hot topic of current cardiovascular research pointing to emerging need to translate knowledge on the long non-coding RNAs associated with cardiovascular disease to clinical praxis for individual patient´s benefit. Paper brings important and original view on the given topic and could be published in NcRNA journal after minor revisions.

*We thank the reviewer sincerely for their time and remarks and will heed their advice for minor revision, along with checking again for typographical errors and improvements.

Comments:

There is quite a high number of typing errors in the manuscript, thus paper should be carefully checked for them, e.g.: Line 4, author´s name “Azod i” should be corrected to “Azodi”

*We think this might have been a formatting error. We have checked now that it does not appear in the final pdf.

Lines 22-28, bigger font size used as compared to the rest of manuscript

*We have rectified this.

Line 62, thete is double “Up to up to”

*We have deleted one of the ‘up to’s

Line 75 “remodelling” replace by “remodeling” as US English is used in whole manuscript (including word “remodeling” in other parts of paper)

*Thank you. Amended.

Line 95 “LnRNAs” replace by “LncRNAs” *Done.

Line 99 “cardomyocyte” replace by “cardiomyocyte” *Done

Line 106 “Vascular” should not start with capital letter “V” *Done

Line 106 “electrophysiologial” replace by “electrophysiological” *Done

Lines 107, 117, please unify “%” or “percent” *Thank you. We have decided on %

Line 118 insert space between “titin” and “which” *Done

Line 136 “utilised” replace by “utilized” (US English) *Done

Line 148 “analyse” replace by “analyze” (US English) *Done

Line 186 insert space between “defect.” and “One” *Done

Abbreviations: please, re-check if all abbreviations are properly explained when first time occur in the text (e.g. BRG1, MeCP2, HDAC, PARP, CAD, SNP…etc. are not explained). Alternativelly you may create the List of abbreviations and insert into the paper.

Finally, please always use abbreviations when once were intorduced, e.g. use “CVD” instead of “cardiovascular disease” in lines 68, 221.

*We have changed a number of the terms “cardiovascular disease” later in the text now to “CVD.”

I suggest to add explaining text to the Figure 1, such as “Original dogma of MB expected that only proteins encoded by genes (exons) exert biological functions, but nowadays it is evidenced that also RNA transcripts/molecules encoded by introns, particularly miRNAs and lncRNAs exert important biological functions, especially regulatory roles in cell physiology” or something similar to make the Figure self-explaining without reading whole paper.

*Thank you for this suggested text. We have included a slightly modified sentence to this effect. Now Lines 53-55.

Check the text for fluent English and English grammar, especially in Future directions some sentences are not clear in meaning, e.g.: Line 198, first sentence, subject is missing: WHO?? is currently available to identify…? Maybe WE??. Alternatively, exclude words “With the” and start sentence with “Different methods…” Line 205, the same as previous – exclude “With the.” or add the subject to the sentence 

*We have amended the Future perspectives section as advised.

Reviewer 2 Report

In the commentary “The missing ‘lnc’ between genetics and cardiac disease,” the authors address the importance of lncRNA dysregulation in Cardiovascular Disease and encourage the community to consider the yet largely unexplored applications of non-coding RNAs in clinical genetics. Some minor comments to improve the manuscript:

The commentary is overall interesting, but several paragraphs need references (ex. paragraphs 47-51 and 52-56). The authors use a lot of hard data throughout the manuscript, but often the source of the facts and numbers is not referenced adequately.

In my opinion, the manuscript contains too many abbreviations. I would suggest to include full names rather than abbreviations, particularly when they are mentioned only once (ex. ET-1/AngII in line 91, polycomb repressor complex 2 (PRC2) in line 93, and TTN in paragraph 117-120). Also, abbreviations should be consistent throughout the text (ex. In paragraph 82-89, we have three forms of MALAT1, Malat-1, and Malat1).

According to the authors, Figure 1 depicts the revised central dogma in molecular biology. However, I find the figure only contains the classical view of transcription (DNA->RNA) followed by translation (RNA->protein). This figure then is entirely out of context with the purpose of the commentary, which is to highlight the critical place in molecular biology for non-coding RNAs that are transcribed but not translated. My suggestion is to modify this figure to include lncRNAs.

Finally, it would be interesting to know the opinion of the authors regarding the reasons why lncRNA inclusion and advancement in clinical genetics is currently lagging.

Author Response

*our responses have been marked with an asterisk

In the commentary “The missing ‘lnc’ between genetics and cardiac disease,” the authors address the importance of lncRNA dysregulation in Cardiovascular Disease and encourage the community to consider the yet largely unexplored applications of non-coding RNAs in clinical genetics. Some minor comments to improve the manuscript:

*We thank the reviewers for their time in reviewing our comment and in the advice given to make in suitable for publication.

The commentary is overall interesting, but several paragraphs need references (ex. paragraphs 47-51 and 52-56). The authors use a lot of hard data throughout the manuscript, but often the source of the facts and numbers is not referenced adequately.

*We have added references to the paragraphs indicated and revised the remainder of our reference list.

In my opinion, the manuscript contains too many abbreviations. I would suggest to include full names rather than abbreviations, particularly when they are mentioned only once (ex. ET-1/AngII in line 91, polycomb repressor complex 2 (PRC2) in line 93, and TTN in paragraph 117-120). Also, abbreviations should be consistent throughout the text (ex. In paragraph 82-89, we have three forms of MALAT1, Malat-1, and Malat1).

*Thank you. We have amended the text so that any abbreviations rarely used are written in full instead.

*Regarding MALAT-1, we have amended all 3 versions to be hyphenated. The higher case is the human form of the lncRNA and the lower case refers to the rodent form, as in the reference and study described.

According to the authors, Figure 1 depicts the revised central dogma in molecular biology. However, I find the figure only contains the classical view of transcription (DNA->RNA) followed by translation (RNA->protein). This figure then is entirely out of context with the purpose of the commentary, which is to highlight the critical place in molecular biology for non-coding RNAs that are transcribed but not translated. My suggestion is to modify this figure to include lncRNAs.

*Thank you for pointing this out and we acknowledge a lack of clarity in that lncRNAs make one the the two functional biological products of the genome. We have now added a label for LncRNAs as one of the functional outputs of transcription.

Finally, it would be interesting to know the opinion of the authors regarding the reasons why lncRNA inclusion and advancement in clinical genetics is currently lagging.

*We have added a sentence to summarise the reasons why lncRNA inclusion and advancement in clinical genetics is currently lagging (lines 201-224)

*The chronological lagging behind in revealing the existence of lncRNAs as important biological mediators, their functional characterization and lack of widespread education and knowledge transfer of their significance to the current medical community all contribute to the slow progress in incorporating lncRNAs in mainstream clinical genetic analyses.